The effect of ribosomal protein S15a in lung adenocarcinoma

Zhang Yifan 1
Zhang Guangxin 1
Li Xin 2
Li Bingjin 2 bingjinli@hotmail.com
Zhang Xingyi 1 xyzhang@jlu.edu.cn
1 Department of Thoracic Surgery, The Second Hospital of Jilin University , Changchun, Jilin , China
2 Jilin provincial key laboratory on molecular and chemical genetic, The Second Hospital of Jilin University , Changchun , China
Yuan Tifei
Electronic publication date: 2016 Mar 14
Publication date: 2016
Volume: 4
Electronic Location ID: e1792
Received 2015 Dec 29; Accepted 2016 Feb 20
Copyright: © 2016 Zhang et al.
Copyright year: 2016
Copyright holder: Zhang et al.
License: This is an open access article distributed under the terms of the Creative Commons Attribution License, which permits unrestricted use, distribution, reproduction and adaptation in any medium and for any purpose provided that it is properly attributed. For attribution, the original author(s), title, publication source (PeerJ) and either DOI or URL of the article must be cited.
License URL: https://creativecommons.org/licenses/by/4.0/

Keywords: shRNA, RPS15A, Tissue microarray, P53 pathway, Lung adenocarcinoma

Funding: Natural Science Foundation of China #81272472 This study was supported by grants from the Natural Science Foundation of China (#81272472). The funders had no role in study design, data collection and analysis, decision to publish, or preparation of the manuscript.

==============================
Background: RPS15A (Ribosomal Protein S15A) promotes mRNA/ribosome interactions in translation. It is critical for the process of eukaryotic protein biosynthesis. Recently, aberrantly expressed RPS15A was found in the hepatitis virus and in malignant tumors. However, the role of RPS15A has not been fully revealed on the development of lung cancer. Method: In this study, a Tissue Microarray (TMA) of primary lung adenocarcinoma tissue specimens was carried out. Furthermore, to further investigate the function of RPS15A in lung cancer, RPS15A-specific short hairpin RNA (shRNA) expressing lentivirus (Lv-shRPS15A) was constructed and used to infect H1299 and A549 cells. Result: Our data showed that RPS15A expression was increased in tumor tissues. Furthermore, the knockdown of RSP15A inhibited cancer cell growth and induced apoptosis in the cancer cells. Gene expression profile microarray also revealed that the P53 signaling pathway was activated in Lv-shRPS15A-infected cancer cells. Conclusion: Taken together, our results demonstrate that RPS15A is a novel oncogene in non-small cell lung cancer and may be a potential therapeutic target in lung cancer.

Introduction

Lung cancer is the most prevalent of malignant tumors and the leading cause of cancer-related death in the world (Jemal et al., 2011). The principal histological forms of lung cancer have been well established. Approximately 85% of lung cancer cases are Non-Small Cell Lung Cancer (NSCLC). Adenocarcinoma is the most commonly diagnosed type (Butnor & Beasley, 2007). Despite advances in surgical techniques and other therapeutic strategies, most patients diagnosed with lung cancer ultimately succumb to this disease within 5 years (Miller, 2005; Paleari et al., 2008). Therefore, comprehensive understanding of the molecular mechanisms underlying lung cancer progress is important for the development of optimal anti-cancer therapy (Hensing et al., 2014).

Tumorigenesis involves multistep development to acquire certain malignant capabilities, such as sustaining proliferative signaling, resisting cell apoptosis, activating invasion and metastasis et al. (Hanahan & Weinberg, 2011). Underlying these abilities of tumor cells, rapid de novo biosynthesis of functional ribosome is essential for cancer cells to aggressively grow and obtain multiple malignant phenotypes. Ribosomes are composed of diverse ribosomal RNAs in eukaryotic organs. So far, most ribosomal proteins have been identified to bind to certain regions of ribosomal RNA and perform catalytic functions. Many ribosomal proteins have various extra-ribosomal functions, such as DNA repair, transformation, development, apoptosis, and transcription (Lee et al., 2010; Nishiura et al., 2013; Nosrati, Kapoor & Kumar, 2014). Some studies show that these proteins may play a role in human tumor development and progression. For example, depletion of ribosomal protein L26 and L29 suppress the proliferation of human pancreatic cancer PANC-1 cells (Li et al., 2012), while RPL22 expression is highly associated with NSCLC (Yang et al., 2013).

Human Ribosomal Protein S15a (RPS15A) is a highly conserved cellular gene that maps to human chromosome 16p12.3 locus (Chan et al., 1994; Schaap et al., 1995). It promotes mRNA/ribosome binding in translation via the interactions with the cap-binding subunit of eukaryotic initiation factor 4F (eIF-4F) (Linder & Prat, 1990). In yeast, G1/S cell cycle phase arrest induced by cdc33 (encoding eIF-4F in yeast) mutation could be reversed by RPS15A over-expression, these findings suggest that RPS15A may play a role in cell cycle transition (Lavoie et al., 1994). RPS15A over-expression also facilitates hepatocellular growth via promoting cell cycle transition and accelerates tumor formation in vitro (Lian et al., 2004), whereas RPS15A mRNA down-regulation inhibited hepatic cancer cell growth (Xu et al., 2014). However, the role of RPS15A in lung cancer has not been completely studied.

The present study was aimed to investigate whether RPS15A is involved in the development and progression of lung cancer. A tissue microarray was carried out to determine the expression of RPS15A. To examine its functional role in lung cancer progress, an RPS15A-specific small interfering RNA (siRNA)-lentiviral vector was constructed to block RPS15A expression in human lung adenocarcinoma H1299 and A549 cells. Furthermore, the impacts of RPS15A silencing on the growth of the cancer cells were examined by MTT assay and colony formation assay. In addition, the effects of RPS15A knockdown on the apoptosis of H1299 and A549 cells were determined by flow cytometry analysis. To further explore the potential molecular mechanisms, we also performed a human whole genome oligo microarray followed by a KEGG pathway enrichment analysis.

Materials and Methods

Tissue microarray and immunohistochemical staining

Tumor samples were collected from 75 patients with lung adenocarcinoma at the Department of Thoracic Surgery, The Second Affiliated Hospital of Jilin University, China, from July 2005 to December 2011. All the tissue samples were obtained from surgery with informed consent and with institutional review board approval of the hospital. Non-tumor samples from the macroscopic tumor margin were isolated at the same time and used as the matched adjacent non-neoplastic tissues. Expression of RPS15A protein was detected using immunohistochemical analysis on commercially available Tissue Microarrays (TMAs) from Shanghai Zhuoli Biotechnology Co., Ltd. (Shanghai, China), which contained a total of 150 tissue samples of tumor or adjacent normal tissues from 75 patients. There were 44 male patients and 31 female patients aging from 32 to 80 years, with an average age of 58 years old. The tumors were classified according to the Tumor Nodes Metastasis (TNM) stage revised by the International Union Against Cancer in 2002.

Immunohistochemical staining and scoring

To detect RPS15A expression in lung cancer, archival paraffin-embedded tumor samples were used to build up tissue microarray (TMA) blocks for Immunohistochemical (IHC) staining. Immunohistochemical staining was performed using the Vectastain Elite ABC Kit (Vector Laboratories, Burlingame, CA, USA) according to the manufacturer’s protocol. Briefly, TMA sections were deparaffinized and hydrated in xylene, ethanol and water. After heat-induced antigen retrieval procedures, sections were incubated overnight at 4 °C with anti-RPS15A primary antibody (1:50 dilution; Abcam, Cambridge, MA, USA). After the primary antibody was washed off, the ABC detection system was performed by using biotinylated anti-rabbit IgG. The slides were counterstained with haematoxylin and mounted in xylene mounting medium for examination. Negative controls were treated identically but with the primary antibody omitted. Three researchers evaluated immunoreactivity independently. The percentage of positive tumor cells was determined by each observer, and the average of three scores was calculated. The proportion of immunopositive cells were categorized as following: intensity of staining: none (0), mild (1), moderate (2), strong (3); percentage of the positive staining: 0 (−), <15% (+), 15–50% (++), >50% (+++). To obtain final statistical results, − and + groups were considered as negative.

Cell lines

Lung adenocarcinoma cell lines H1299 and A549, lung squamous cancer cell line SK-MES-1, as well as small cell lung cancer cell line H1688 (Cell Bank of Chinese Academy of Sciences, Shanghai, China) and Human Embryonic Kidney (HEK) 293T cell line (American Type Culture Collection, ATCC, Manassas, VA, USA) were maintained in DMEM (Hyclone, Logan, UT, USA) with 10% FBS (Hyclone) and penicillin/streptomycin at 37 °C in humidified atmosphere of 5% CO2.

Construction and infection of RPS15A short hairpin (shRNA)-expressing lentivirus

To permit robust inducible RNAi-mediated RPS15A silencing in tumor cells, RPS15A-specific shRNA containing lentiviral vector was constructed. The RNAi was designed based on conservative cDNA fragments within the coding region of human RPS15A gene (NM_001019)-targeting sequence (5′-GCAACTCAAAGACCTGGAA-3′) of oligo nucleotides. The sequences were annealed and ligated into the Age I/EcoR I (NEB, Ipswich, MA, USA)-linearized pGCSIL-GFP vector (Shanghai Genechem Co. LTD., Shanghai, China). The lentiviral-based shRNA-expressing vectors were confirmed by DNA sequencing. Recombinant lentiviral vectors and packaging vectors were then cotransfected into 293T cells using Lipofectamine 2000 (Invitrogen, Carlsbad, CA, USA), according to the manufacturer’s instructions for the generation of recombinant lentiviruses Lv-shRPS15A and negative control Lv-shCon. The culture supernatants containing lentiviral particles expressing Lv-shRPS15A and Lv-shCon were harvested and ultra-centrifuged 48 h after transfection, respectively. H1299 and A549 cells were infected with the lentiviruses at Multiplicity of Infection (MOI) of 10 and 20, respectively.

Quantitative real-time PCR analysis

In brief, total RNA was extracted using TRIzol reagent (Invitrogen, Carlsbad, CA, USA). The reverse transcription reactions were carried out following the protocol of the M-MLV Reverse Transcriptase (Promega Corp., Madison, WI, USA). Real-time quantitative PCR analysis was performed using SYBR Master Mixture Kit (TaKaRa, Dalian, China). The primer sequences for PCR amplification of RPS15A were 5′-CTCCAAAGTCATCGTCCGGTT-3′ and 5′-TGAGTTGCACGTCAAATCTGG-3′. GAPDH was applied as an internal control. The primer sequences of GAPDH were 5′-TGACTTCAACAGCGACACCCA-3′ and 5′-CACCCTGTTGCTGTAGCCAAA-3′. 2−ΔΔCT method was adopted to calculate the relative expression levels of RPS15A by subtracting CT values of the control gene from the CT values of RPS15A.

MTT proliferation and colony formation assay

Briefly, exponentially growing cancer cells were inoculated into 96-well plates with 2 × 104 cells per well. After incubation for 24, 48, 72, 96 and 120 h, 10 μl of sterile MTT (5 mg/ml) was added into each well. Following incubation at 37 °C for 4 h, the reaction was blocked by adding 100 μl of dimethyl sulfoxide. The formazan production was determined by measurement of the spectrometric absorbance at 490 nm. The values obtained are proportional to the amount of viable cells and each experiment was repeated three times. In colony formation assay, H1299 and A549 cells infected with Lv-shRPS15A or Lv-shCon were seeded in six-well plates with 5 × 102 cells per well and cultured at 37 °C with 5% CO2 for 14 days. The cell colonies were washed twice with PBS, fixed in 4% paraformaldehyde for 30 min and stained with Giemsa for 20 min. Individual colonies with more than 50 cells were counted under a fluorescence microscope.

Apoptosis assay by Fluorescence-Activated Cell Sorting (FACS) analysis

In apoptotic cell detection, exponentially growing H1299 and A549 cells were seeded in six-well plates. After 48 h, cells were collected and washed with pre-chilled PBS (4 °C). Then, the cells were centrifuged at 1500 rpm for 5 min. After discarding the supernatant, the pellet was resuspended with binding buffer. The cells were then incubated with 5 μl Annexin V-APC for 15 min in the dark. After fitration of the cell suspension, the analysis of apoptotic cells was performed by FACS can (Becton–Dickinson, Franklin Lakes, NJ, USA).

Gene expression profile microarray

Briefly, A549 cells were seeded in a six-well plate at a density of 4 × 105 cells per well and infected by Lv-shRPS15A and Lv-shCon at MOI of 20, respectively. After 96 h, the total RNAs were extracted using TRIzol reagent (Invitrogen, Carlsbad, CA, USA) and were used for cDNA synthesis and labeling, microarray hybridization and then followed by flour-labeled cDNA hybridizing their complements on the chip (Affymetrix, Santa Clara, CA, USA). The resulting localized concentrations of fluorescent molecules were detected and quantified by GeneChip Scanner 3000 (Affymetrix). Finally, the data were analyzed by Expression Console Software (Affymetrix) with default RMA parameters. Data are representative of three separate assays.

Gene ontology annotation and KEGG pathway enrichment analysis

The gene ontology analysis was performed to functionally annotate the differentially expressed genes according to the Gene Ontology database (http://www.geneontology.org/). The pathway enrichment analysis was performed according to the KEGG (Kyoto Encyclopedia of Genes and Genomes) database. The Fisher’s exact test and χ2 test were applied to classify the significant GO categories and pathways, and the FDR was calculated to correct the P-value by multiple comparison tests. A P-value <0.05 and an FDG <0.05 were set as thresholds to select significant GO categories and pathways.

Western Blotting assay

In brief, A549 cells were collected and lysed with precooled lysis buffer after 96 h of infection. Total protein was extracted from the cells and determined by the BCA method. Protein (20 μg) was loaded onto a 10% SDS-PAGE gel. The gel was run at 30 mA for 2 h and transferred to poly-vinylidene difluoride membrane (Millipore, Billerica, MA, USA). The resulting membrane was blocked in 5% non-fat dry milk blocking buffer and then probed with rabbit anti-P21 (1:2,000 dilution; Abcam, Cambridge, MA, USA; Cat. ab7960), mouse anti-TP53I3 (1:500 dilution; Abcam, Cambridge, MA; Cat. ab123917), rabbit anti-SESN2 (1:500 dilution; Abcam, Cambridge, MA, USA; Cat. ab57810) and mouse anti-GAPDH (1:6,000; Santa Cruz Biotechnology, Inc., Sana Cruz, CA, USA) overnight at 4 °C. The protein level of GAPDH was used as a control and detected by an anti-GAPDH antibody. The membrane was washed three times with Tris-Buffered Saline Tween-20 (TBST), followed by incubation for 2 h with anti-rabbit and anti-mouse IgG at a 1:5,000 dilution (Santa Cruz Biotechnology, Inc., Santa Cruz, CA, USA). The membrane was developed using enhanced chemiluminescence (Amersham, Little Chalfont, UK). Bands on the developed films were quantified with an ImageQuant densitometric scanner (Molecular Dynamics, Sunnyvale, CA, USA).

Statistic analysis

Data were expressed as mean ± SD. Comparisons were performed by two-sided independent Student’s test, one-way ANOVA analysis and χ2 test using SPSS software for Windows version 23.0 (SPSS, Chicago, IL, USA). Kaplan-Meier survival curves were plotted and log rank test was done. Statistical significance was accepted when P < 0.05. All experiments carried out in this study were repeated three independent times.

Result

RPS15A was significantly overexpressed in lung adenocarcinoma tissues

We detected the expression of RPS15A protein in a tissue microarray (TMA) of primary lung adenocarcinoma and adjacent normal lung tissue specimens using immunohistochemical staining with an anti-RPS15A antibody. RPS15A immunostaining was primarily detected in cytoplasm. Representative examples of RPS15A protein expression in lung cancer and normal lung samples are shown in Fig. 1A. The immunopositive rates of RPS15A in lung adenocarcinoma and normal lung tissues were 66.7% (50/75) and 42.7% (32/75), indicating that RPS15A was highly expressed in lung cancer in comparison with adjacent normal tissues, as shown in Table 1 (P < 0.001). The correlation between RPS15A expression and different clinical pathological factors in lung adenocarcinoma is shown in Table 2. No significant correlation was found between RPS15A expression and age, grade, TNM stage, tumor size and lymph node metastasis (P < 0.05). Kaplan-Meier survival analysis of overall prognosis between patients with higher RPS15A expression and patients with low RPS15A expression was carried out. No significant correlation between RPS15A expression and overall prognosis was found, as shown in Fig. 1B.

Figure 1 Immunostaining of RPS15A with tissue microarray.

Immunostaining of RPS15A in lung adenocarcinoma and adjacent normal tissues with tissue microarray. (A) Three representative cases with different expression status of RPS15A, ranging from negative, mild and strong expression were taken at 100 × and 400 × magnification in lung cancer and normal tissues. (B) Kaplan-Meier survival analysis of overall prognosis between patients with higher RPS15A expression and patients with low RPS15A expression. Log-rank test was used to statistically calculate the difference.

Table 1 RPS15A expression in 75 lung adenocarcinoma and adjacent normal tissue specimens.

Histological types	Number	RPS15A expression	P*	
−	+	++	+++		
Cancer	75	11	19	15	35	0.000	
Normal tissue	75	5	38	26	6		
Note:

* P values were obtained with the χ2 test, P < 0.001.

Table 2 Correlation between RPS15A expression and clinicopathological factors in 75 lung adenocarcinoma patients specimens.

Variables	All patients	RPS15A expression	P*	
Negative	Positive		
Total	75	26	49		
Age(y)					
 ≤60	48	14	25	1.000	
 >60	27	12	24		
Gender					
 Male	43	19	24	0.053	
 Female	32	7	25		
TNM stage					
 I∼II	47	20	36	0.788	
 III∼IV	28	6	13		
Tumor size					
 ≤3 cm	8	1	7	0.249	
 >3 cm	67	25	42		
Lymph node metastasis					
 Yes	28	8	20	0.458	
 No	47	18	29		
Note:

* P values were obtained with the χ2 test.

Efficacy of lentivirus-mediated RNAi targeting RPS15A

Quantitative real-time PCR was performed to detect the fundamental expression of RPS15 mRNA in several lung cancer cell lines, including lung adenocarcinoma cell lines H1299, A549 and lung squamous cancer cell line SK-MES-1, as well as small cell lung cancer cell line H1688. As shown in Fig. 2A, RPS15A expression levels were quite obvious in the cancer cell lines. Consequently, loss of function assay was applied through RPS15A knockdown. H1299 and A549 cell lines were selected for subsequent studies. To knockdown RPS15A expression, H1299 and A549 cells were infected by the lentiviruses stably expressing RPS15A-specific shRNA (Lv-shRPS15A). Lentivirus expressing negative shRNA (Lv-shCon) was used as negative control. More than 80% of GFP-expressing cells were observed under fluorescence microscope after 72 h (Fig. 2B). The silencing effect of lentivirus mediated RPS15A RNAi on RPS15A expression in H1299 and A549 cells was examined through real-time PCR and Western Blotting assay. The expression level of RPS15A in the Lv-shRPS15A infected cells was significantly lower than that in the Lv-shCon infected cells (Figs. 2C and 2D). Therefore, these data indicated the high efficacy of lentivirus mediated RPS15A silence in lung cancer cells.

Figure 2 Lentiviral mediated RPS15A downregulation.

(A) Relative RPS15A mRNA level in lung cancer cell lines (H1299, A549, SK-MES-1, H1688 and H1975). (B) Fluorescence photomicrographs of H1299 and A549 cells 72 h after lentivirus infection. (C, D) RPS15A mRNA and protein expressions were dramatically downregulated in Lv-shRPS15A infected cells evidenced by real-time PCR and Western blotting assay. **P < 0.01 versus Lv-shCon.

RPS15A silence inhibited NSCLC cell growth in vitro

To investigate the role of RPS15A in the proliferation of lung cancer cells, the proliferative abilities of Lv-shCon and Lv-shRPS15A infected H1299 and A549 cells was determined by MTT assay. During the 120 h incubation period, the growth of Lv-shRPS15A infected cells was significantly slower than that of Lv-shCon infected cells at the time points of 48 h, 72 h, 96 h and 120 h (Figs. 3A and 3B). To investigate the effect of RPS15A downregulation on tumor formation, colony formation assay was performed. Quantitative analysis of colonies showed that after incubation for 10 days, the number of colonies in Lv-shRPS15A infected cells was significantly lower than that in the Lv-shCon infected cells (Figs. 3C–3F). Therefore, the low colony-forming efficiency of Lv-shRPS15A infected H1299 and A549 cells demonstrated that RPS15A silencing inhibited the colony forming ability of lung cancer cells in vitro.

Figure 3 The proliferation of H1299 and A549 cells.

The proliferation of H1299 and A549 cells was inhibited after Lv-shRPS15A infection determined by MTT assay. (A, B) The colony formation abilities of H1299 and A549 cells were determined by colony formation assay after Lv-shRPS15A infection. (C, D) Images of colonies and statistical analysis of the number of colonies. (E, F) Images of colonies recorded under microscope. *P < 0.05, **P < 0.01 versus Lv-shCon.

RPS15A silence triggered cell apoptosis in vitro

To determine whether RPS15A knockdown induces apoptosis in H1299 and A549 cells, Annexin V-APC staining was performed and the percentage of apoptotic cells was detected by flow cytometry. As shown in Figs. 4A–4D, Lv-shRPS15A infected cells exhibited significantly higher proportion of apoptotic cells than that of Lv-shCon infected cells, especially in Lv-shRPS15A infected A549 cells. This suggested that downregulation of RPS15A expression might trigger apoptosis in lung cancer cells, which contributed to the cell growth suppression.

Figure 4 RPS15A knockdown induced apoptotic cells.

RPS15A knockdown induced apoptotic cells were determined by flow cytometry analysis after Annexin V-APC staining. (A, B) Histograms of FACS analysis. (C, D) Percentage of apoptotic cells. **P < 0.01 versus Lv-shCon. (E) Key factors of P53 signaling pathway, such as P21, TP53I3 and SESN2, were examined in Lv-shRPS15A infected A549 cells by using western blotting method. The protein level of GAPDH was employed as a control.

RPS15A knockdown activated P53 signaling pathway in vitro

To explore the potential downstream mechanisms underlying the actions of RPS15A silencing, a cDNA microarray assay was performed to compare the differential gene expression profiles between Lv-shRPS15A infected and Lv-shCON infected A549 cancer cells. Microarray analysis demonstrated that 885 genes were upregulated and 566 genes were downregulated significantly in Lv-shRPS15A infected cancer cells when compared to Lv-shCON infected cells. To further classify the function of these genes, GO analysis was carried out. The most significant GO categories of biological processes included protein, biopolymer and cellular macromolecule metabolic processes. Of these genes, the most enriched GO in terms of molecular function involved DNA binding, cation and ion binding and enzyme regulator activity. KEGG pathway enrichment analysis revealed that certain signaling pathways participated in the tumor inhibition induced by RPS15A knockdown, of which P53 signaling pathway members were most evidently annotated. The top 20 significantly perturbed pathways are listed in Table 3. To confirm the P53 pathway activation, the expressions of key factors of P53 signaling pathway were determined through western blotting. Consistent with microarray results, P21 and TP53I3 expressions were upregulated by RPS15A knockdown, while SESN2 expression was downregulated, as shown in Fig. 4E.

Table 3 KEGG pathway enrichment analysis revealed that P53 signaling pathway members were most evidently annotated.

Top 20 significantly perturbed pathways are listed.

	

Discussion

Therapeutic drugs targeting cancer-related molecules can specifically inhibit malignant cells, causing minimal adverse off target reactions because of the well-defined mechanisms (Chen et al., 2013). Therefore, identification of molecular factors responsible for pulmonary carcinogenesis and elucidation of their underlying mechanisms of proliferation are urgently needed for novel therapeutic targets.

Earlier genetic studies in zebrafish suggested that many ribosomal protein genes are haploid-sufficient suppressor gene that are highly related to tumorigenesis (Amsterdam et al., 2004). Ribosomal proteins are abundant in most cells and well known for their functional role in the assembly of ribosomal subunits at the early stages of translation. In addition, ribosomal proteins have been found to regulate cell proliferation, apoptosis, DNA repair and gene transcription. Currently, ribosomal proteins are emerging as novel regulators of cancer cell growth, whose mutations and changes of expression level are highly relevant to human malignancies. RPS15A, as a component of the 40S ribosomal subunit, has been found to facilitate the binding of capped mRNA to the ribosomal subunit 40S in translation initiation. Furthermore, it has been reported that downregulation of its mRNA expression in hepatocellular carcinoma cells significantly inhibits tumor growth, suggesting that RPS15A may play a role in human carcinogenesis. Yet, up to now, little is known about its function in NSCLC cells.

To determine the expression levels of RPS15A in lung adenocarcinoma tissues, TMAs was performed and revealed that RPS15A was highly expressed in lung cancer tissue. Therefore, we hypothesized that RPS15A may play an important role in the proliferation of lung cancer. In this regard, a lentivirus-mediated RNAi system was applied to inhibit RPS15A mRNA expression in human lung adenocarcinoma H1299 and A549 cells in vitro. Lentivirus expressing RPS15A-specific shRNA was constructed and used to infect H1299 and A549 cells. The efficiency of lentivirus-induced silencing of endogenous RPS15A was confirmed by qPCR and western blotting assay. To determine the impact of RPS15A knockdown on the lung cancer growth in vitro, an MTT assay and colony formation assay were carried out. As a result, downregulation of RPS15A expression greatly impaired the proliferation and colony-forming ability of H1299 and A549 cells. Furthermore, flow cytometry analysis data showed that RPS15A silencing induced apoptosis as characterized by the prominent presence of apoptotic cancer cells.

To elucidate the downstream mechanisms underlying RPS15A silence in lung cancer, we carried out a human whole genome oligo microarray and KEGG pathway enrichment analysis. The data revealed that the P53 signaling pathway was activated significantly in Lv-shRPS15A infected A549 cells. The P53 pathway has been well known for its anticancer function through initiating apoptosis, cell cycle arrest, maintaining genomic stability, angiogenesis inhibition etc (Levine & Oren, 2009). To confirm the P53 signaling activation, we determined the expression of P21, TP53I3 and SESN2 by using western blotting. P21(P21), which is tightly controlled by the tumor suppressor protein p53, is a potent cyclin-dependent kinase inhibitor that binds to cyclin-CDK2 or -CDK4 complexes, thus functioning as a regulator of cell cycle progression at G1 (el-Deiry et al., 1993). TP53I3 is induced by the tumor suppressor p53 and is thought to be involved in p53-mediated cell death (Polyak et al., 1997). SESN2 is also known for its function in the regulation of cell growth and survival in cellular response to different stress conditions (Hay, 2008).

Ribosomal proteins are often classified as cell growth associated molecules due to their key role in protein synthesis (Eid et al., 2014; Lian et al., 2004; Wang et al., 2014). Our finding that RPS15A downregulation inhibits NSCLC cell growth is also supported by previous studies that RPS15A knockdown also inhibits hepatic cancer cell growth (Xu et al., 2014). Like other ribosomal proteins, RPS15A has been found to involve extra-ribosomal functions in cell growth, apoptosis and cell cycle transition. In this study, our findings that RPS15A induced apoptosis and cell cycle phase arrest in lung cancer A549 cells further supported that P53 signaling is critical for the regulation of apoptosis and cell cycle transition. However, to date, the issue of whether and how RPS15A interacts with other regulators remains poorly studied, and further investigation is warranted to elucidate the detailed mechanisms underlying the action of RPS15A. Taken together, our study demonstrated that RPS15A might serve as an upstream modulator of P53 signaling pathway.

In conclusion, RPS15A expression was increased in tumor tissues. Furthermore, the knockdown of RSP15A inhibited cancer cell growth and induced apoptosis in the cancer cells. Gene expression profile microarray also revealed that the P53 signaling pathway was activated in Lv-shRPS15A-infected cancer cells. Therefore, our findings demonstrate that RPS15A is a novel oncogene in non-small cell lung cancer and may be a potential therapeutic target in lung cancer.

Supplemental Information

Supplemental Information 1 Raw data of manuscript.

Raw data-The effect of ribosomal protein S15a in lung adenocarcinoma.

Click here for additional data file.

Additional Information and Declarations

Competing Interests

Author Contributions

Data Deposition

Bingjin Li and Xingyi Zhang are Academic Editors for PeerJ.

Yifan Zhang conceived and designed the experiments, performed the experiments, wrote the paper, prepared figures and/or tables, reviewed drafts of the paper.

Guangxin Zhang conceived and designed the experiments, analyzed the data, contributed reagents/materials/analysis tools.

Xin Li conceived and designed the experiments, prepared figures and/or tables.

Bingjin Li conceived and designed the experiments, analyzed the data, wrote the paper, prepared figures and/or tables, reviewed drafts of the paper.

Xingyi Zhang conceived and designed the experiments, analyzed the data, contributed reagents/materials/analysis tools, wrote the paper, prepared figures and/or tables, reviewed drafts of the paper.

The following information was supplied regarding data availability:

Raw data can be found in the Supplemental Information.

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
