# Peer review of "The effect of ribosomal protein S15a in lung adenocarcinoma"

_PeerJ, doi:10.7717/peerj.1792_

## Round 0.1 · original submission · Major Revisions

· Academic Editor

Major Revisions

Please check the comments from the two reviewers and make necessary revisions as requested.

Reviewer 1 ·

Basic reporting

1. Line 261, where are the Figure 2E and 2F?
2. Line 301, Table 2 is the table 3.
3. There are numerous grammatical errors throughout the text. For example, line 236 “Representative examplesof”, there was no blank between “examples” and “of”.

Experimental design

Original primary research is appropriate within scope of the journal.

Validity of the findings

1. In the line 238, the criterion for calculating the immunopositive rates is not consistent. According to Table 1, the total number of cancer samples is not 75, but 80; however, the denominator used for the calculation is 75. In addition, why did authors not considered ‘+’ as immunopositive staining?
2. In the line 250, authors determined the expression of RPS15 mRNA in several lung cancer cell lines, including H1299, A549, H1975, SK-MES-1, and H1688. Authors, however, only selected H1299 and A549 for subsequent studies; such as, in the Figure 2A, there was no expression level of H1975. Authors must provide the reason why they omitted H1975. Furthermore, why did authors choose H1299 in spite of its lowest level of expression?
3. Statistical analysis should be included in Figure 3A and 3B.
4. Authors consistently provided both H1299 and A549 data. So, if possible, please provide key factors of p53 signaling pathway in H1299 cell in Figure 4E, too.

Additional comments

N/A

·

Basic reporting

No Comments

Experimental design

No Comments

Validity of the findings

No Comments

Additional comments

The whole of manuscript was well written, and the study was well designed. However, some issues need to be clarified as follows:
1.In figure 1A, what's the purpose that they detected the expression of RPS15A in these cell lines ? It makes no sense in this manuscript.
2.In the Results, the authors mentioned that RPS15A is obviously expressed in lung cancer cell lines. The authors should compare the expression of RPS15A in lung cancer cells to normal lung cells, in order to suggest whether the RPS15A is overexpressed or underexpressed compared to normal cells. These information can be important in the further clinical application. 
3. In figure 3c, the quality of the colony formation assay is poor. No obvious colonies are showed, and they are not supposed to be concentrated in the central area.
4. The figure legends lack necessary details to fully understand the figures. A more comprehensive figure legends is needed.
5. There is typos and grammatical errors in the manuscript. Overall, the manuscript needs proofreading and polishing of the language.

---

## Round 0.2 · accepted · Accept

· Academic Editor

Accept

Congratulations on this nice work.

Reviewer 1 ·

Basic reporting

No Comments

Experimental design

No Comments

Validity of the findings

No Comments

Additional comments

The authors adequately addressed my concerns/critiques and the manuscript is significantly improved.

·

Basic reporting

No Comments

Experimental design

No Comments

Validity of the findings

No Comments

Additional comments

no